# Incorporating Field Experience into International Agricultural Development Programs

Alexis Zickafoose * and Gary Wingenbach 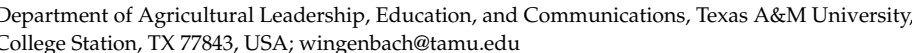

Department of Agricultural Leadership, Education, and Communications, Texas A&M University,
College Station, TX 77843, USA; wingenbach@tamu.edu

*   Correspondence: alexis.zickafoose@ag.tamu.edu

**Abstract:** Some graduate education programs support experiential learning but do not require practical experiences for students specializing in international agriculture development. We examined U.S. land grant university graduate international agricultural education program descriptions for experiential learning requirements and found them lacking. The literature surrounding volunteerism is reviewed and examples are described. International volunteerism can fill the experience gap for future international agricultural development professionals studying at U.S. land grant institutions. Graduate students can acquire practical field experiences through short- or long-term volunteer assignments, such as those in the USAID Farmer-to-Farmer volunteer program for international agriculture development. Graduate students build their capacities by providing technical and agricultural assistance in Farmer-to-Farmer assignments, whereas host country participants enhance their knowledge, skills, and abilities to expand and increase agricultural sector productivity. Short-term international volunteer assignments help graduate students gain practical experience, improve interpersonal skills, and enhance cultural competencies. Host communities and volunteers alike benefit by engaging in intercultural exchanges that promote increased understanding of differing societies worldwide.

**Keywords:** volunteerism; graduate students; agriculture; capacity building

## 1. Introduction

Institutions prioritize hiring people with global perspectives and cross-cultural competence [1,2]. In response, some agricultural student and faculty development programs have increased their internationalization efforts [2–5]. Undergraduate agriculture programs identified international awareness as an education priority. Faculty development programs were reported to spur global mindsets and establish inter-university cooperation among participants [5]. However, many graduate agriculture programs lack internationalization efforts beyond the classroom.

Many U.S. land grant universities' agricultural graduate education programs encourage experiential learning [6], but few have incorporated international practical field experiences into their curricula [7]. International field experiences help graduate students transfer academic theories and principles into real-world situations [7]. By requiring international field experiences during graduate education, students seeking careers in international development may be better prepared to meet career expectations of cultural competence after graduation.

## 2. International Agricultural Development Programs

Several U.S. land grant universities offer international agricultural development-related programs at master's and doctoral program or certificate program levels. These programs aim to prepare students for agriculture-related development careers in academia

and industry [8–17]. Most institutions blend practice, theory, and research into a comprehensive program to help students gain perspective and knowledge of current issues in global agriculture [8–17]. Although many postsecondary programs boast a variety of courses, there is minimal evidence of "required" international experience to augment classroom learning. Few require international experience via research or internship [9,14–16], and fewer require foreign language skills [15,16]. A lack of field experience may inadequately prepare graduate students for careers in international development.

## 3. International Volunteerism

About one million Americans volunteer internationally each year [18]. International volunteering implies working without financial compensation in a foreign country to help people or communities achieve social, economic, or political goals [19,20]. Although international volunteerism started after World War I, more widely recognized organizations did not start until the 1960s and 1970s [19,21]. Governmental and non-governmental volunteer organizations targeted developing countries in Africa, Latin America, Asia, and the Pacific [22–25]. Currently, some organizations working in international development (e.g., Winrock International, United Evangelical Mission, and Weltwärts), recruit volunteers from high-, middle-, and low-income countries to increase social understanding between volunteers and host country nationals, and to build host communities' economic development [19,21,26–29].

Many development organizations' goals align with current United Nations (UN) Sustainable Development Goals (SDGs) [30–32]. The SDGs aim to end poverty, improve health and education, reduce inequality, and increase economic growth while mitigating climate change and preserving natural resources in developed and developing countries [33]. The SDGs correlate with many volunteer organizations' goals and objectives, especially in strengthening host community ownership of development needs and solutions.

Incorporating UN and organizations' development goals in volunteerism programs with graduate education, could address host communities' development needs and augment students' abstract learning with practical experience. This study sought to (1) review characteristics of international volunteerism, (2) examine land grant universities' graduate education programs with an international development focus, and (3) propose mutually beneficial outcomes for host development communities and graduate students. This paper provides an overview of international volunteerism, and the strengths and weaknesses of international agricultural development programs at U.S. land grant universities are examined. Then, the solution of incorporating short-term high-skilled international volunteer experiences into U.S. land grant universities' international agricultural development programs is proposed and discussed.

## 4. International Volunteerism in a Globalized World

Historically, international volunteer programs either promote international understanding or provide development aid and humanitarian relief [24]. However, these ideas are not always separate. Development organizations such as the U.S. Peace Corps, include both through its programs to promote mutual understanding [34,35]. Separating these concepts does occur, as seen in volunteer tourism programs focusing on international understanding [36,37].

The context of international volunteerism is changing. Recruiting young unskilled northerners to volunteer in the global south is no longer accurate; more focus is placed on short-term professional labor, especially in information technology and fundraising [31]. In this new context, volunteers serve to supplement local expertise and improve host organization processes through education rather than direct service [38,39].

International volunteer field assignments take many forms. Volunteers' assignments can vary by skill level, duration, and group vs. individual-based activities. Volunteer experiences range from short-term unskilled assignments to multiyear skilled positions [18,19,40].

Short-term assignments typically are 1–8 weeks, medium-term is 3–6 months, and long-term is 6 or more [23].

An important distinction in assignment type is through group- or individual-based activities. Volunteers' assignments with group-based activities may incorporate work camps, humanitarian responses, and mission trips, where organized groups volunteer together to achieve a goal [40]. Individual-based activities are found in the USAID (United States Agency for International Development) Farmer-to-Farmer program or the U.S. Peace Corps. Peace Corps recruits train as a cohort but rarely work as volunteer groups after placement into a two-year volunteer assignment. Peace Corps volunteers conduct individual projects in different or sometimes the same community and occasionally work in pairs or small groups for community-based projects.

Short-term skilled volunteer assignments may work best with graduate education because of time and cost. Short-term assignments enable graduate students to miss fewer classes during the academic year but increase volunteer opportunities during holidays and/or summer breaks. Graduate students serving short-term assignments provide technical assistance to host communities while improving practical skills. Research shows that short- and long-term skilled volunteers were more effective than less skilled volunteers. Additionally, volunteers with more education and advanced skills were more valued across organizations [40].

Short-term skilled virtual volunteer assignments may also be viable during the academic year. Before the novel coronavirus pandemic of 2019 (COVID-19), some international volunteer organizations incorporated virtual assignments into their volunteer programs, such as those with the United Nations [19,41]. Virtual volunteerism refers to assignments completed through online spaces [41]. The COVID-19 pandemic restricted international travel, which influenced a boom in virtual volunteerism. Several volunteer organizations, such as Partners of the Americas, Winrock International, ACDI/VOCA, and the U.S. Peace Corps, offered virtual volunteer assignments to continue their development efforts during the COVID-19 pandemic.

Virtual international volunteer opportunities increased during the COVID-19 pandemic because development needs increased. Many businesses were forced to pivot, nationally and internationally, to a virtual marketplace. Virtual volunteerism provided organizations and universities with opportunities to assist small businesses in economic crisis so they could transition to a digital marketplace [42,43]. These experiences provided development or humanitarian aid because of the limited ability in creating mutual understanding with a host community.

Virtual volunteerism creates international opportunities for older adults, people with disabilities, and youth [44]. More people can engage in international volunteerism from the comfort of their homes. Not everyone can travel internationally; virtual volunteerism engages community members who otherwise could not participate in onsite assignments. Although some traditional onsite assignments occurred during the pandemic, virtual volunteerism increased organizational capacity, despite travel barriers [44]. Building the virtual volunteerism infrastructure could enhance long-term programming [44]. Some international organizations continue offering virtual volunteer assignments, as seen in volunteer opportunity webpages [35,45,46].

International volunteerism has goals such as improving individual, household, or social standards of living [33,35,47]. However, when volunteers do not uphold organizational goals, their work may negatively impact the host communities' development. Volunteers who do not maintain organizational goals in development may jeopardize such programs and their careers.

Negative impacts on host communities can occur from international volunteer assignments. Bargeman et al. [34] found mixed results on the effectiveness of a volunteer tourism program in Ghana. Volunteer teachers caused local Ghanaian teachers to re-teach subjects because the volunteers did not follow the school curriculum [34]. Research is scarce about the impacts of international volunteerism on host country individuals and communities.

Most research focuses on positive program outcomes and/or volunteers' outcomes, rather than on host community outcomes [23,48].

International volunteerism attracted criticisms, such as reinforcing colonial and imperial ideologies [19,23,31,49–52]. Volunteerism can become neocolonialistic when international volunteers and their organizations do not focus on empowering host country nationals' sense of self-sufficiency or when volunteer programs primarily benefit the volunteer's personal development [19,23,31]. Inadequately prepared volunteers may reinforce negative stereotypes, contributing to 'us' vs. 'them' mentalities, such as perceptions of the global South being dependent on the global North [50,52,53]. International volunteerism that empowers host country nationals' sense of self-sufficiency and host communities' sustainability (i.e., economic, environmental, and social) helps minimize the neocolonial aspects of volunteerism. Host communities gain ownership of development programs when they are included in project design and management processes. Collaborative, rather than individual efforts, create opportunities for mutual understanding and respect.

Volunteerism may improve one's intercultural understanding, personal development, and community development; however, international volunteerism may also replace local labor [54,55], have negative health outcomes [56,57], and/or hinder host community growth [54,58]. These criticisms are intervention-specific and stem from systemic program design and implementation issues rather than from individual practices.

Successful volunteerism occurs when international organizations and volunteers understand host community issues and find creative ways to support them [59]. International volunteers gain a plethora of knowledge, skills, and abilities during international volunteer assignments. They develop interpersonal skills, learn or practice new languages, develop cross-cultural competencies, contribute to a global society, and positively change their attitudes about foreign cultures and international social and economic development [21,23,36,60–62]. After volunteers return home, they report more employment possibilities and increased life satisfaction [63–65]. Graduate students can benefit from broadened global perspectives and interpersonal skill development from international volunteer assignments. International volunteerism builds graduate students' knowledge about host communities, knowledge that may not be acquired in university settings. Foreign host community's indigenous knowledge systems, culture, lifestyles, etc., can supplement graduate students' curricula and provide realistic settings to apply theoretical information acquired in traditional university settings.

Because possible negative impacts exist in some assignments, international volunteer programs should be vetted before graduate students engage with such opportunities. This is key in planning international volunteer assignments that benefit graduate students and host communities alike. The Farmer-to-Farmer model only operates in communities that express the desire for a volunteer to assist the hosts with a particular skill for a short-term assignment. Farmer-to-Farmer's administering organizations have in-depth knowledge of the culture and context in which they work. They leverage volunteers' expertise to respond to the needs of local communities and prepare volunteers for their service by providing local context information. The high-skilled short-term volunteer model can enable the appropriate delivery of technical knowledge and skills desired by the host community.

## 5. Strengths of International Agricultural Development Graduate Programs

Some graduate programs with international agricultural development foci aim to prepare students for a globalized world [8–17]. Among these 10 U.S. land grant universities, eight offer master's degrees, three have doctoral programs, and four offer graduate certificates or minors in international agriculture development (Table 1).

Many graduate-level courses focused on international agricultural development exist in U.S. land grant universities. These courses were analyzed by topic/theme; 28 emerged (Table 2). Most common were those in research methods and policy. 'General elective' courses were specified as required for degree completion. Some courses targeted international agricultural development while others were more generic development. The

'other' category included courses in tourism, culture, food safety, and food science. Overall, graduate programs with international agricultural development courses included many topics, although a consistent focus was not shared in all programs. "Research" was seen as important across most programs. Much of graduate education concerns research conduct, which may be learned in the classroom or through practice by conducting research projects with faculty members.

**Table 1.** International agricultural development programs at U.S. land grant universities.

| Universities | Degree(s) | Program Titles |
| --- | --- | --- |
| Cornell | Master of professional studies (MPS) | Global Development |
| The Ohio State University | MS/Ph.D. | Agricultural Communications, Education, and Leadership-International Agriculture |
| Oklahoma State University | M.Agr./MS | International Agriculture |
| Penn State University | Dual-title MS/Ph.D. | International Agriculture and Development |
| Rutgers | MBS | Master of Business and Science- Global Agriculture |
| Texas A&M University | Certificate | International Agricultural Development |
| Texas A&M University | M.Agr. | International Agriculture |
| Texas A&M University | MS/Ph.D. | Agricultural Leadership, Education, and Communications-International Agricultural Development |
| University of California, Davis | MS | International Agricultural Development |
| University of Georgia | Certificate | International Agriculture |
| University of Maryland | Certificate | International Agriculture and Natural Resources |
| University of Missouri | Graduate Minor | International Development |

**Table 2.** Graduate-level international agriculture course foci.

| Theme | Courses | Number of Universities |
| --- | --- | --- |
| Research methods | 34 | 6 |
| Policy | 18 | 4 |
| General Electives | 16 | 7 |
| Development | 16 | 8 |
| Education | 14 | 4 |
| Economics | 13 | 4 |
| International Agricultural Development | 12 | 6 |
| Environment | 10 | 3 |
| Communications | 9 | 4 |
| Organizations | 8 | 4 |
| Business | 8 | 3 |
| Experience | 7 | 4 |
| Farming | 7 | 4 |
| Leadership | 7 | 2 |
| Food system | 6 | 3 |
| Other | 6 | 3 |
| Food security | 5 | 3 |
| Sociology | 4 | 2 |
| Ethics | 3 | 2 |
| Gender class race | 3 | 2 |
| Sustainability | 3 | 2 |
| Animal Science | 3 | 1 |
| Foreign language | 2 | 2 |

The most common course topic across most universities ($n = 8$) was development, international agricultural development courses were available at six universities. As eight and six universities had at least one course on development and international agricultural development, respectively, there seems to be agreement on the need for courses that prepare students for practical development experiences, especially in agriculture. However, one

would expect that all programs that claim to prepare students for international development and agricultural development courses in their curriculum.

## 6. Weaknesses of International Agricultural Development Graduate Programs

Among the 10 U.S. land grant universities' programs studied, only 6 offered a course specifically on international agricultural development (Table 2). The finding was surprising because some degrees were titled international agricultural development. Four universities offered or required courses involving practical field experiences such as travel/study abroad or internships. Despite claims these universities prepared graduate students for an increasingly globalized world, there appeared to be a disconnect between program content and outcome. Practical experience in international development cannot be delivered in a classroom. Without practical experience, professionals in international development can lack information about context and culture and their importance when creating and administering agricultural development programs abroad. Concrete experience with differing cultures and contexts can increase cultural understanding and lead to adjustment and adaptation [66,67]. U.S. graduate programs in international agricultural development that do not provide practical experience are not adequately preparing their graduates for careers in development.

Graduate programs with an international agricultural development focus should emphasize the importance of research and fieldwork in international settings. Coursework in international agriculture may provide fundamental knowledge, but culture shapes how some research and programs are viewed and/or conducted in many countries. International volunteer opportunities could supplement graduate students' international development coursework with practical field experience during graduate studies.

## 7. Incorporating Field Experience into International Agricultural Development Programs

Graduate programs with an international agricultural development focus aim to develop students' knowledge, skills, and abilities in agriculture through research and study of global food and fiber issues. However, practical experiences outside the classroom appear lacking in many programs. International service may increase cross-cultural awareness among volunteers and develop host communities' social, health, and economic development. We believe that increasing student opportunities for practical field experience can be leveraged through international volunteer short-term assignments (USAID Farmer-to-Farmer program).

The Farmer-to-Farmer program provides immense opportunities for volunteers and host communities to benefit from its short-term assignments. Many volunteers work in social and health improvement, children's programs, economic and agricultural modernization, and/or civil society development programs [21]. Exchanges between volunteers and host country nationals highlight the importance of local and global engagement [19,23,68]. Graduate students could provide technical assistance (i.e., skills and knowledge), increase collaborative and community-based research, gain cultural understanding, promote empowerment, strengthen networks, and build capacity during short-term volunteer assignments [19,23,59,69,70].

The Farmer-to-Farmer model provides a unique experience for both host communities and their volunteers. Funded by USAID, a variety of multinational non-governmental organizations administer the program. Those organizations provide support to both the community and the volunteer through appropriate skills matching. By the organizations ensuring that the volunteer is qualified for the short-term position they are filling, the community knows that they are receiving a professional and highly skilled individual. Farmer-to-Farmer also removes the burden of providing for the volunteer during their term from the community. By supporting the volunteer, Farmer-to-Farmer enables the host community to focus on the work they and the volunteer will do. These short-term

assignments are also available virtually. As Farmer-to-Farmer already has the structure in place for both in-person and virtual assignments, international agricultural development graduate students could work on projects that fit their skills while classes are in session or over extended breaks. Volunteers are supported by Farmer-to-Farmer during their host country assignments, where they gain insights about international development through a multinational non-governmental organization. If U.S. land-grant universities encouraged their graduate students to participate in Farmer-to-Farmer assignments, the students would gain field experience to propel them into their future careers.

Graduate education programs with international agricultural development foci can be enhanced by supplementing curricula with practical international field experiences. Both authors are former Peace Corps volunteers (Republic of Georgia and Guatemala), and both completed multiple short-term Farmer-to-Farmer volunteer assignments (Colombia, Nigeria, Ghana, Bangladesh, and the Dominican Republic). Whereas these recommendations represent the culmination of shared experiences, they are based on one author's more recent international volunteer assignment experiences.

During my recent graduate studies, I participated in several Farmer-to-Farmer assignments. My Peace Corps experience influenced my decision to seek a graduate program focused on international agricultural development. Peace Corps and Farmer-to-Farmer programs helped develop my interpersonal and technical skills; both experiences afforded me opportunities to witness changes in behavior at the individual and host community levels. These programs fostered my language, technical, research, and interpersonal skills. While serving as an international volunteer, I wholeheartedly state that I learned as much as I taught host country nationals. Whereas my long- and short-term experiences had differing scopes and purposes, my knowledge, skills, and abilities were refined through those experiences. Peace Corps and Farmer-to-Farmer experiences helped me broaden my global perspectives. Additionally, I witnessed the effects of working collaboratively with host country nationals to build capacity in agricultural sector development.

I participated in two Farmer-to-Farmer virtual assignments during the COVID-19 pandemic. I could not practice language skills in virtual assignments, but I learned about local customs and trained host communities on digital media marketing. I used skills learned in my graduate education and adapted them to the needs of the communities with whom I was working. Virtual volunteer opportunities allow graduate students to continue their education while engaging in development work. Short-term volunteer assignments afforded me opportunities to apply academic knowledge to practical international agricultural development settings. U.S. land-grant universities, in partnership with Farmer-to-Farmer or other short-term volunteer programs, could provide graduate students and others (e.g., county extension agents, non-profit agency members) with international experiences to build capacity among volunteers and host community participants alike. Graduate students seeking careers in international agricultural development should acquire field-based, short-term experiences in volunteer assignments before engaging in long-term international development positions.

## 8. Conclusions

Graduate students in international development lack concrete experiences which prevents their acquisition of nuanced knowledge and skills that should be applied during international development interventions. Much information across agricultural education, plant breeding, and animal science is context-dependent. Preparing students to work in academia and industry requires a holistic education that builds theoretical and practical skills. Short-term skilled in-person and virtual volunteer opportunities could fill graduate students' international experience gap. International volunteerism can build individuals' technical and interpersonal skills while sustainably benefiting communities. By engaging with host communities, graduate students can supplement their classroom education making them more adequately prepared for long-term international development positions.

**Author Contributions:** Conceptualization, A.Z. and G.W.; methodology, A.Z. and G.W.; formal analysis, A.Z.; investigation, A.Z.; data curation, A.Z.; writing—original draft preparation, A.Z. and G.W.; writing—review and editing, A.Z. and G.W.; visualization, A.Z.; supervision, G.W.; project administration, A.Z. and G.W.; All authors have read and agreed to the published version of the manuscript.

**Funding:** This research received no external funding.

**Institutional Review Board Statement:** Not applicable.

**Informed Consent Statement:** Not applicable.

**Data Availability Statement:** No new data were created or analyzed in this study.

**Conflicts of Interest:** The authors declare no conflict of interest.

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
