# Peer review of "Incorporating Field Experience into International Agricultural Development Programs"

_education, doi:10.3390/educsci13050456_

Round 1

Reviewer 1 Report

The main concerns are around structure of the paper. Some suggestions:

The abstract says nothing about the content of the paper. Your paper is an argument for experiential learning in a specific program, international agricultural development. Perhaps changing the second sentence to: We examined land grand university international agricultural program descriptions for requirements related to direct international experience and found it lacking. The literature surrounding volunteerism is reviewed and examples are described. The USAID Farmer-to- Farmer... 

Introduction - why not tell the reader what you will do in the next paragraphs. The paragraphs do not flow easily.

The section about the Program websites seems out of place. I would put that after the literature review of International Volunteerism. 

I would move most of the Discussion into this review of International Volunteerism - pros and cons of these programs. Since this is an Opinion piece, the discussion is the body of the article. 

A clear paragraph about why the Farmer-to-Farmer model is different and desirable. 

Then, describe the Graduate Programs and why their lacks of courses on Sustainabilty and requirements for real world practical experience is glaring.

The Discussion section should only talk about why the Farmer-to-Farmer program is the answer to the problem that you have identified.

Conclusion paragraph was strong and read well

Reviewer 2 Report

An interesting and useful analysis, but with one major drawback - it is limited exclusively to data and examples from the USA. My strict recommendation is to analyze and present norms, globalization and innovations all over the world, even superficially outside the USA. The title itself contains the term Globalization, but the article conceived in this way looks more like "Americanization" than Globalization.

I have no other comments.

Round 2

Reviewer 1 Report

Excellent revision. Article is clearer and better understood. 

Reviewer 2 Report

-